# Self-Healing Hydrogels: Development, Biomedical Applications, and Challenges

**DOI:** 10.3390/polym14214539

**Published:** 2022-10-26

**Authors:** Md. Mahamudul Hasan Rumon, Anwarul Azim Akib, Fahmida Sultana, Md. Moniruzzaman, Mahruba Sultana Niloy, Md Salman Shakil, Chanchal Kumar Roy

**Affiliations:** 1Department of Chemistry, Bangladesh University of Engineering and Technology (BUET), Dhaka 1000, Bangladesh; 2Department of Textile Engineering, Northern University, Dhaka 1208, Bangladesh; 3Department of Mathematics and Natural Sciences, BRAC University, Dhaka 1212, Bangladesh; 4Department of Biochemistry and Molecular Biology, Jahangirnagar University, Dhaka 1342, Bangladesh; 5Department of Pharmacology & Toxicology, University of Otago, Dunedin 9016, New Zealand

**Keywords:** hydrogel, mechanical properties, modulus, self-healing, tissue engineering, drug delivery

## Abstract

Polymeric hydrogels have drawn considerable attention as a biomedical material for their unique mechanical and chemical properties, which are very similar to natural tissues. Among the conventional hydrogel materials, self-healing hydrogels (SHH) are showing their promise in biomedical applications in tissue engineering, wound healing, and drug delivery. Additionally, their responses can be controlled via external stimuli (e.g., pH, temperature, pressure, or radiation). Identifying a suitable combination of viscous and elastic materials, lipophilicity and biocompatibility are crucial challenges in the development of SHH. Furthermore, the trade-off relation between the healing performance and the mechanical toughness also limits their real-time applications. Additionally, short-term and long-term effects of many SHH in the in vivo model are yet to be reported. This review will discuss the mechanism of various SHH, their recent advancements, and their challenges in tissue engineering, wound healing, and drug delivery.

## 1. Introduction

Hydrogels are considered one of the most attractive materials for biomedical applications due to their similarities in mechanical and chemical behavior to natural tissues [1,2]. The water retention capacity of hydrogels is very high in their integrated polymer network [3]. Hydrogels can be prepared using natural polymers, e.g., agarose, alginate, chitosan, collagen, fibrin, gelatin, etc. [4,5], while they can also be made of synthetical synthetic polymers or a combination of natural and synthetic polymers. Hydrogels exhibit mechanical and dimensional responses to the changes in their surrounding environment, such as pH, temperature, and ionic strength [6]. These behaviors represent their potential applicability for drug delivery systems, biomedical adhesives, sensors, tissue reconstructions, fabrication of artificial organs, and more [7,8,9]. The self-healing hydrogels (SHH) are hydrogels that show special healing ability. The self-healing property of hydrogel comes from the reversible physical or chemical bonds [10]. Furthermore, SHH also show ionic conductivity and adhesive properties that make them suitable for real-time applications including tissue engineering, cartilage treatment, wound healing treatment, and so on [11]. Moreover, SHH should process enough mechanical toughness to be used as a biomedical material [12].

The development of polymeric hydrogels has become ubiquitous over the last couple of decades [13]. To address the rising demand, researchers are tuning hydrogel quality by changing the synthetic process and bonding mechanisms to introduce mechanical robustness and other properties [9]. High-end mechanical behavior, long-elongation, strain-hardening, and self-healing capability are the significant characteristics of fabricating hydrogel to use as artificial tissue [14]. Most conventional polymeric gels have demonstrated considerable mechanical properties with poor adhesion and self-healing ability, which have restricted their use in sensitive applications such as tissue engineering and drug delivery [15]. However, these properties are difficult to bring together in the same hydrogel effectively [15].

The enhancement of mechanical behavior often produces weak self-healing performance [16]. Assuring mechanical toughness and optimum self-healing performance is a tricky job in SHH development. Efforts have been made to design synthetic hydrogels to provide the desired and unique combination of high mechanical properties with strong self-healing behavior by introducing special types of functional groups to provide various reversible chemical networks of hydrogels, such as hydrogen bonds, ionic bonds, hydrophobic interactions, host–guest interactions, π–π stacking, sometimes special covalent bonds, etc. [17] The most effective strategy for developing tough hydrogels has been observed in polyampholyte hydrogels by introducing a supramolecular system [18]. 

The Young’s modulus of a gel composite depends on the elastic properties, and this depends on the numbers of both reversible and nonreversible covalent bond active sides present on the composite matrix [19]. Self-healing ability is an intrinsic property, and this particular property of hydrogel either comes from the reversible physical or chemical bonds or the combination of those bonds [20]. Viscous properties of SHH may reduce the modulus of composite hydrogel [21]. As a result, the molecular interactions can happen properly for the viscous region, whereas a hydrogel with a large modulus requires strong bonding with moderate viscous properties [10]. This may reduce the chances of the bond regeneration process and lead to a tough gel with weak healing ability. However, to introduce a mechanically tough but self-healable hydrogel, the proper combination of elastic and viscous properties or viscoelastic property is necessary [16,22]. Recently, Rumon et al. [16] reported the viscoelastic property of their prepared material by the proper combination both of reversible and nonreversible chemical and physical active sides simultaneously.

Building on the concept of the reversible dynamic bond to make healable hydrogels, the scientific community has already introduced several nanoparticles, including graphene sheets and their derivatives such as graphene oxide, carbon nanotube, various metal nanoparticles, etc. [12,23]. These materials provide either dynamic reversible physical or reversible chemical bonds. However, dynamic reversible bonds, including hydrogen bond interactions, π-π stacking interactions, Van der Waals bonding, ionic bonds, etc., and the reversible covalent bonds have been continuously directed to design self-healable but tough hydrogels [16,24]. The biggest weakness of those strategies described is that they cannot address the root cause of the opposite relationship between mechanical toughness and self-healing ability. Dahlquist criterion indicates that as the polymeric materials’ modulus increases, the self-healing ability is decreased, and high self-healing efficiency is difficult to obtain in large modulus hydrogels [16,18,25]. For example, Fan et al. [26] reported a conventional hydrogel where tannic acid was used as a crosslinker to crosslink polyacrylamide and polyvinyl alcohol monomer [26]. Though this gel exhibited considerable toughness, it showed a very nominal level of healing ability that was only 40%. A self-healing polyampholyte gel was synthesized by Ihsan et al. [18], of which it was described that, in the increment of the chemical crosslinker, the modulus is also enhanced, resulting in a substantial lower self-healing efficiency from 92.11% to 2.76% [18].

Detailed discussions on the trade-off relation between mechanical toughness and self-healing behavior were not reported in recent reviews [27,28,29]. In the current narrative review, we included research articles published before April 2022 that covered our topics of interest (i.e., mechanical toughness and SHH). “Hydrogel”; “mechanical properties”; “Young’s modulus”; “self-healing”; “tissue engineering”; “wound healing”; and “drug delivery” keywords were searched in the article title or abstract from four scientific databases (Google Scholar, Scopus, Web of Science, and PubMed).

Many review articles already reported various bonding mechanisms for self-healing performance [10,12,17,29,30]. However, the trade-off relation between self-healing and mechanical properties is yet to be reported and scrutinized. In this review article, we will discuss the mechanisms of self-healing and relative mechanical properties of hydrogels and their applications in tissue engineering and drug delivery.

## 2. Self-Healing Mechanism in Hydrogels

Self-healing hydrogel is an intrinsic polymer, and this special healing ability of hydrogel either comes from the reversible physical or chemical bonds or a combination of those bonds [31]. Other properties of SHH are conductivity, fast adhesion, and stimuli-responsiveness [32,33,34]. More importantly, the biomedical application of SHH should possess enough mechanical toughness [10].

The hydrogel self-healing mechanism possesses similarities with biological systems [35,36]. The healing process progresses through five consecutive individual steps: surface rearrangement, surface approach, wetting, diffusion, and randomization [37,38]. All those steps are facilitated through the molecular interaction of two fractured surfaces, which leads to regeneration or reconstruction of the damaged bonding to ensure the healing process. However, they contains only reversible and dynamic physical and chemical bonds [39].

### 2.1. Self-Healing Hydrogels with Physical Bonds

The role of physical bond chemistry on the healing ability or healing performance based on reversible non-covalent interactions such hydrogen bonds, hydrophobic interactions, host–guest bond interactions, and ionic or electrostatic interactions is all included in this section. Different self-healing hydrogels show different bonding dynamics. Figure 1 shows the bonding mechanism of functionalized graphene oxide crosslinked polyacrylamide hydrogel [16]. 

The hydrogen bond is an electrostatic attraction force among hydrogen atoms and electronegative atoms such as oxygen, nitrogen, and fluorine that is covalently bonded with a hydrogen atom [40,41]. It is a dynamic and reversible bond that has a vital impact on developing the SHH system [42,43]. For example, Zhang et al. [44] reported polyvinyl alcohol-based hydrogels with exceptional self-healing performance when the polymer concentration was around 36% wt. The hydroxyl groups present in the polymer matrix promote hydrogen bond reformation and make the material self-healable [44]. The donor and acceptor conformations of these hydrogen bonds also considerably influence hydrogen bond formation and affect self-healing behavior. Rumon et al. [16] reported a functionalized graphene oxide-based polyacrylamide gel that provides about 70% healing efficiency. The graphene oxide surface has a large abundance of -OH and COOH groups, which undergo physical bond formation with the polymer chain. As a result, the hydrogel provided up to 70% self-healing ability. In addition to this work, Rumon, with his team, showed the effect of crosslinking on healing kinetics, and reported that, as the degree of crosslinking increased, the graphene oxide based crosslinker (GOBC) crosslinked polyacrylamide (PAM) hydrogels showed faster healing phenomena. Figure 2 shows the stress-strain curve and the healing kinetics of PAM hydrogel with various GOBC crosslinkers [16]. In this study, the authors showed that the percentage of the healing efficiency increases as time passes, and the highest GOBC containing PAM composite hydrogel possesses maximum and fast healing, while the MBA cross-linked gel composite does not undergo any healing process.

The conformational structure of the donors and acceptors of hydrogen bonds can affect the strength of the bonds, which can also affect how well the self-healing process works [45]. Zimmerman et al. [46] proposed that healing efficiency strongly depends on the position or arrangement of lone pair electron donors and acceptor groups, and he demonstrated the mechanism using just alphabetic letters (A and D). Here, A is the lone pair electron and D is the lone pair electron donor. The AA-DDD array has a larger binding constant than the DAA-AAD and ADA-DAD arrangements. A self-healing hydrogel was developed by Zhang et al. [47] with quadruple hydrogen bonds by using ureido pyrimidinone (UPy). It is possible to undergo dimerization of the UPy moieties by using those quadruple hydrogen bonds. Multiple hydrogen bonds in the hydrogel network enabled the self-healing process [47]. Ye et al. [48] developed guanosine and cytosine (nitrogen base)-based SHH that provide self-healing through reversible hydrogen bond interactions among the nitrogen atoms on the guanosine and cytosine and the hydrogen atoms.

Despite their self-healing ability, hydrogels based on hydrogen bonding are less stable in water. To make them stable in an aqueous environment [49], particular focus has been given to hydrophobic binding interactions. In polymeric gels, hydrophobic monomers are introduced as co-monomers with hydrophilic monomers to prepare a hydrophobic bond-based self-healing hydrogel [50]. This hydrophobic monomer-based hydrogel provides self-healing performance because of the hydrophobic chain’s free energy and entropy gain. Hydrophobic polymer chains always try to keep away from the aqueous atmosphere of a swollen hydrogel and subsequently retain surface-bound water molecules from the hydrogel matrix [51]. More importantly, hydrogels undergo a self-healing process through entropy or free energy gain, which results in entropy change. For instance, Tuncaboylu et al. [52] and Deng et al. [53] developed several hydrogels based on hydrophobic interactions, and most of those gels exhibited satisfactory self-healing performance. Owusu-Nkwantabisah et al. [54] prepared self-repairing gels from a hydrophilic monomer’s poly(N-isopropyl-acrylamide). The self-healing properties of hydrophilic hydrogels could be affected by temperature, and healing efficacy increases as the temperature rises [54]. Mihajlovic et al. [55] and Tuncaboylu et al. [52] prepared a self-healing hydrophobic micellar hydrogel by mixing a hydrophilic monomer (acrylamide) and a hydrophobic monomer (stearyl methacrylate sodium dodecyl sulfate). In all those cases, adding a small quantity of salt to the stearyl methacrylate sodium dodecyl sulfate aqueous solution may lead to the micelle formation and solubilization of hydrophobes within the stearyl methacrylate sodium dodecyl sulfate micelles. Additionally, stearyl methacrylate sodium dodecyl sulfate micelles in the matrix with time-dependent dynamic moduli exhibited large strain values with better self-healing properties (Table 1) [52]. In contrast, in time-independent dynamic moduli, hydrogels demonstrated substantial mechanical properties but no self-healing ability.

Most traditional synthetic self-healable hydrogels based on hydrophobic interaction are usually brittle and have limited mechanical features [70,71]. This drawback can severely restrict their applications, as they often need high mechanical toughness and load-bearing ability. However, host–guest interaction-based supramolecular hydrogels have shown a prominent approach to overcoming this drawback. In host–guest interactions, more than two molecules form a complex using dynamic non-covalent interactions such as hydrogen bonding, van der Waals attraction forces, and electrostatic or coordination bonds [72]. Moreover, this interaction involves a particular assembly among two compounds, like a lock and key by inclusion thanks to their complementary structures. Therefore, through dynamic reversible and non-covalent bond interactions, a guest molecule takes place inside of a host molecule of a second species [73]. The reversible behavior of the interactions in this assembly provides a dynamic character to the crosslinking formed in this way that has been used in the self-healing processes [74]. Crown ether, cyclodextrin [75], and cucurbituril are typical examples of host molecules that are mainly responsible for self-healing properties [76,77,78]. In addition, adamantine, cholic acid, ferrocene, or azobenzene are typical guest molecules accommodated by various guest species with different binding affinities, making it a stable insertion complex with CDs and their derivatives [76,79]. Deng et al. [80] synthesized several conductive hydrogels that exhibited self-healing properties. These hydrogels could be used for biomedical applications where electroactivity is needed, as biosensors, as reported by Deng et al. [80], and as a carrier for therapeutic medium or agent, as reported by Chen et al. [81]. The typical mechanism of the host–guest interaction-based redox-active self-healing hydrogel system is shown in Figure 3. 

In a recent study, electrostatically attractive oppositely charged ions crosslinked hydrogels have received focus because of their high mechanical properties [82]. These electrostatic attractions among the oppositely charged polyions in the hydrogel matrix make it self-healable using the ionic mechanism. Moreover, external conditions such as the degree of ionization of the weak polyelectrolytes, salt conditions, etc., may influence this straightforward ionic mechanism for self-healing [83]. The migration velocity of free ions and free polyelectrolytes may also affect these electrostatic network properties. It can be influenced by the molecular weight of electrolytes or ions. Dynamic ionic bonding between anionic monomers such as poly(acrylic acid) and ferric ions has been employed to prepare mechanically tough hydrogels; these bonding networks have exhibited the ability to be recovered after damage [84]. The catechol moiety and its derivatives show coordinate bonding with metal ions such as ferric, calcium, or cupric ions. The possible mechanism of metal ion coordinate self-healing hydrogel is illustrated in Figure 4.

### 2.2. Self-Healing Hydrogels with Chemical Bonds

Most mechanically tough polymeric gels have been synthesized by using irreversible covalent interactions [72]. However, reversible crosslinked covalent bonds in dynamic chemistry can provide the self-healing character [85]. In the chemically self-healing hydrogel system, damaged or cracked polymer chain networks can regenerate their bonding interaction through reversible covalent bonds such as disulfide bond, dynamic imine bond, and Diels-Alder reaction.

The imine (or Schiff base) is a molecule containing a double bond of carbon–nitrogen developed when an amine reacts with an aldehyde or ketone nucleophilically [86]. They can be aromatic or aliphatic. Many SHH have been developed using these bases [87,88,89]. The aromatic Schiff base-based hydrogel demonstrates higher mechanical toughness than aliphatic Schiff bases [90,91]. Zhang et al. [86] have reported dynamic imine bond-based self-healing gels for biological applications. Most of the time, benzaldehyde-functionalized polyethylene-glycol (PEG)-based hydrogels were synthesized with aromatic Schiff bases. To prepare those hydrogels, the Schiff-base DF-PEG solution was mixed with either synthetic polymeric materials such as glycol chitosan solution or natural polymer solution such as cellulose solution, depending on their properties and potential uses (Table 2). Moreover, these self-healable hydrogels are also applied in various biomedical systems, such as three-dimensional cell culture, regulated biological molecular release, blood capillary development, CNS recovery, and so on [88]. A pH-dependent self-healable hydrogel has been designed by Guo et al. [92]. They demonstrated that the self-healing Schiff base-based hydrogels are dynamic and their efficiency is strongly influenced by solution pH, as the phenylboronic ester complex has a pH effect on the complex formation, as shown in Figure 5. According to this research, the pH level requirement must be met to achieve sufficient healing efficiency to be used in biomedical applications.

Fu et al. [105] developed an imine and acylhydrazone from a dynamic bond Schiff- base based hydrogel system that has considerable self-healing in which sodium alginate was used to crosslink the N-carboxyethyl chitosan and adipic acid. They demonstrated excellent bond stability in self-healing. Injectable SHH were introduced by Yesilyurt et al. [106]. They are composed of four-armed PEG terminated with phenylboronic acid, and four-armed PEG terminated with diol bonds. According to that study, the pKa value of the phenylboronic acid group is strongly linked to hydrogels’ self-healing properties and mechanical properties [106]. The hydrogel was hard and brittle when prepared at a pH that was just higher than the pKa of the phenylboronic acid group, and the capacity to self-heal was observed to decrease as the pH was raised. The hydrogels demonstrated good self-healing abilities when they were prepared close to the pKa value of the phenylboronic acid group. Gel was not formed in the pH below the pKa value. Moreover, the conformation of the diol groups in the polymer chains can also affect the self-healing process of the phenylboronic ester complex. For example, polysaccharides and phenylboronic acid were used to investigate the binding constants of Schiff base in different diol positions or conformations by Yesilyurt et al. [106]. He noted that phenylboronic acid prefers cis-diol groups over trans-diol groups. Although boron-cis-diol complexation is more thermodynamically strong and likely to repair after disintegrating, the hydrogel’s self-healing capacity is expected to improve when cis-diol is incorporated into polymers used to produce the hydrogel network. The phenylboronic ester complexation-based self-healing hydrogel SHH can be used in clinical applications. Kotsuchibashi et al. [81,107] prepared a series of SHH. Those gels can be used for glucose sensing, medication release, and 3D cell growth.

Like imine bond-based self-healing hydrogel systems, dynamic reversible disulfide bond-based covalent chemistry strongly depends on pH and redox potential [108,109]. The dynamic reversibility of the disulfide bonds can be explained in terms of disulfide/thiol exchange reactions. Zhang et al. [110] designed a hydrogel that can heal spontaneously using a tri-block copolymer (ABC). This polymer contains dithiolane groups in the form of an A-block, and these groups are crosslinked using reversible ring-opening polymerization, which is mediated by disulfide exchange reactions between 1,2-dithiolanes and dithiols. When subjected to external stimuli such as changes in pH and temperature, the hydrogel that has been prepared goes through a reversible sol-gel transition [110]. The as-prepared polymer hydrogel restored itself (around eight times the strain value) after several rheological deformations at 25 °C and was reported as a promising candidate for drug delivery. However, the disulfide exchange of 1,2-dithiolane can reconstruct their chain networks in a solution that has a pH of 7 or higher than 7, and the temperature of the surrounding environment can further regulate the degree of reformation. Tu et al. [42] proposed a special type of hydrogel that shows moderate self-healing performance by using both disulfide bond and imine bonding mechanisms. The polyethylene oxide (PEO) hydrogels exhibited variable self-healing efficiency at different pH because these two dynamic covalent reversible bonds work at different pH conditions (the pKa value significantly influences these two complexation processes’ dynamic systems that further influence the self-healing ability) [42]. Within this PEO hydrogel matrix, the disulfide bond exchange reactions took place at basic pH, acylhydrazone exchange reactions occurred at low pH or acidic pH, and these covalent bonds were not capable of reforming at neutral pH.

Temperature-sensitive Diels-Alder reaction is another dynamic covalent bond chemistry in SHH [111]. The Diels-Alder reaction is a ring-opening reaction between a diene and a dienophile [112]. However, the Diels-Alder reaction has limited biomedical uses since it requires high temperatures and extended times to break and reform. However, the Diels-Alder has recently become popular in polymeric hydrogels to improve its medical applications by combining with other reversible bonds, including the acylhydrazone bond, imine bond, coordination, and electrostatic interaction [72].

## 3. Applications

SHH are one the promising materials for biomedical applications because of their similarities with natural tissues [113,114,115]. The tendency of natural tissues to self-heal is an intriguing ability which extends their lifespan and improves their strength and durability [116]. The very close and similar properties of self-healing hydrogels with natural tissue lead to their real-time applications, including soft robotics, artificial tissue engineering, drug delivery, carrier systems for transporting various guest molecular inside the cell, catalyst chemistry, and also in the electrochemical applications including sensing technology.

### 3.1. Tissue Engineering

Self-healing is an intriguing ability of natural tissues to extend their lifespan and improve their strength and durability [116]. Inspired by biological systems, self-healing properties make hydrogels one of the most promising materials in biomedical engineering [117]. This is because hydrogels have high water content and almost similar mechanical properties to natural tissues [118]. Hydrogels have fast adhesion properties, stimuli-responsiveness, and high conductivity that can be profoundly used, more specifically in tissue engineering [119]. In Figure 6 the various potential applications of SSH are illustrated.

Myocardial infarction, commonly known as a heart attack, occurs when one or more regions of the heart muscle do not receive adequate oxygen. To control myocardial infarction, host–guest type SHH are often used [120]. Agrawal et al. [121] developed a host–guest-based SHH system incorporating pathogenic endothelial cells (PECs) into the hydrogel and inserted them into myocardial tissues [121]. The hydrogel was composed of adamantine- and *β*-CD-modified hyaluronic acid (HA). It should be noted that *β*-CDs-HA were used as hosts and adamantine-HA were used as guests. A mouse myocardial infarction model was used to verify the hydrogel encapsulating PECs. The synergistic effect resulted in a massive improvement in vasculogenesis compared to the PECs and the HA-based hydrogel alone.

An injectable shear-thinning hydrogel system was developed by Loebel et al. [122] via host–guest interaction and Michael addition to deliver minimally invasive therapy to infarcted myocardium to limit remodeling of the left ventricule. The hydrogel was composed of adamantane/thiol modified HA and CD/methacrylate modified HA, where the CD/methacrylate was used as the host, and the adamantane/thiol was used as the guest. Shear slimming injection and high retention were given by a reversible host–guest relationship and the stable addition of Michael [123,124]. During the study of myocardial infarction in rats, epicardial injection of hydrogel led to a significant change in the untreated community and hydrogel without the inclusion of Michael [122,123]. Recently, a self-healing hydrogel containing ureidopyrimidinone (UPy) groups with four hydrogen bonds functionalized by poly(ethylene glycol) PEG were developed as growth factor injection vectors [125]. A pocket was implanted under the kidney capsule of rats with a UPy-modified PEG hydrogel containing an antifibrotic growth factor. The number of myofibroblasts in the contralateral (healthy) kidney remained unchanged after the injection of growth factor-containing hydrogels but increased dramatically after the injection of saline or hydrogel alone [126,127]. Another study used a UPy-modified PEG hydrogel to transmit growth factors to the infarcted myocardium to fix it [127]. This pH-switchable hydrogel could have been inserted via the catheter mapping system’s long and narrow lumen and quickly formed a hydrogel in contact with tissue. In a pig model of myocardial infarction, the growth factor-containing hydrogel diminished scar collagen [128]. 

To repair the central nervous system (CNS), SHH based on glycol chitosan and di-functionalized poly (ethylene glycol) (GC-DP) were designed by Tseng et al. [89]. Injection of GC-DP hydrogel mixed with neurospheres into a CNS-affected zebrafish (Danio rerio) sample facilitated functional regeneration. Moreover, neurosphere-like progenitors showed more significant proliferation and differentiation in GC-DP hydrogel. In another study, the GC-DP hydrogel was combined with an optogenetic method by a temporal-spatial approach to treat neurodegenerative diseases [129]. The hydrogel containing bacteriorhodopsin plasmid and neural stem cells was injected into CNS-impaired zebra fish, which showed signs of neural recovery, particularly when exposed to green light. Furthermore, GC-DP hydrogel was used to stimulate the development of blood capillaries [129]. A hybrid hydrogel system with an interpenetrating polymer network (i.e., a double network) of GC-DP and fibrin was formed in addition to the fibrin gel [129]. The hydrogel caused vascular endothelial cells to develop capillary-like forms, and injections of the hydrogel alone facilitated angiogenesis in zebrafish and restored blood supply in mice with ischemic hindlimbs.

### 3.2. Wound Treatment

Wound healing is a complex and dynamic process of repairing missing or damaged cellular structures and tissue layers [130,131]. SHH may lessen the need for dressing change, saving patients from experiencing needless discomfort [132,133]. Additionally, certain stimuli-responsive SHH may be dissolved by provoking network disintegrations [134]. Furthermore, SHH can fulfill the requirement for surgical debridement and minimize the discomfort associated with dressing changes for burn patients [131,133].

To repair skin tissues, a PEG-based SHH crosslinked with silver ions (Ag^+^) was designed and crosslinked with mangiferin liposomes (MF-Lip) [135]. After seven days, the PEG-based polymeric gel displayed a progressive and controlled release of about 95% of MF-Lip with 67% trapping efficiency. Additionally, the MF-Lip@PEG hydrogel exhibited a strong cytoprotective activity towards hypoxia-induced apoptosis in HUVEC cells via regulation of the Bax-Bcl-2-caspase-3 signaling pathway. More importantly, the MF-Lip@PEG hydrogel showed anti-infection, anti-inflammation, proneovascularization, and anti-necrotic effects(Figure 7) [135]. Wang et al. [136] synthesized Poly-ε-L-lysine (antimicrobial peptide) and oxidative hyaluronic acid-based SHH (FHE@exo) that release adipose-derived mesenchymal stem cells exosomes in treatment of chronic wound. The FHE@exo hydrogel exhibited multifunctional properties including self-healing, stimuli-responsive exosomes release, injectability, and antibacterial activity. This bioactive SHH significantly improved migration, proliferation, and angiogenesis in HUVEC cells. Additionally, it promoted cellular proliferation and neovascularization leading to collagen remodeling, re-epithelialization, and granulation tissue formation at a faster rate in diabetic mice model [136]. Chouhan et al. [137] developed decorin-releasing SHH eye drops for the treatment of retinal scars using gellan gum and sodium chloride solution. Decorin (leucine rich glycoprotein) is an anti-scarring molecule that lessens fibrotic scarring by sequestering TGF-β1 and TGF-β2. The decorin-loaded SHH showed cytocompatibility against human corneal cells. Furthermore, in ex vivo rat corneal damage model, decorin was released at an incremental rate (45%) within 3 h, resulting in improved reepithelialization [137].

### 3.3. Drug Delivery

Lipophilicity is one of the crucial parameters linked with the cellular uptake and cytotoxicity of drugs, or drug candidates. It is also linked with the pharmacokinetics of drugs or compounds. To be an ideal drug, there must be a suitable balance between hydrophobicity and lipophilicity of drug-like molecules [138]. There are several benefits of using SHH or polymers compared to conventional hydrogels or polymers for drug delivery [38,139]. For example, SHH offers homogeneous encapsulation of conjugated drugs or drug candidates, preventing drug diffusion, improving potency, and selectivity, and reducing side effects. More importantly, the release of the drugs can be controlled using external stimulus, while the sustained release effect assures long-term therapeutic benefit (Figure 8) [140,141,142].

Doxorubicin (DOX) is a water-soluble FDA-approved anticancer drug used to treat many cancers, including breast, lung, thyroid, and hematological malignancies [138,143]. Qu et al. [92] synthesized a polysaccharide-based SHH using N-carboxyethyl chitosan and dibenzaldehyde-terminated poly(ethylene glycol) to deliver DOX in HepG2 hepatocellular carcinoma cells [92]. Interestingly, hydrogels having a dynamic covalent Schiff-base linkage showed rapid self-healing properties without any external stimulus [85]. Moreover, the pH-responsive hydrogels demonstrated pH-dependent gel degradation and DOX release in the in vitro system [92,144]. The cytotoxicity studies confirmed the DOX-loaded hydrogel’s concentration and time-dependent cytotoxic activities toward HepG2 cells [145,146,147]. More importantly, the DOX-loaded hydrogel has superior or equivalent cytotoxic activity compared to free DOX and excellent biocompatibility in L929 mouse fibroblasts cells, indicating the promise of DOX-loaded hydrogel in cancer therapy [92,148].

One of the major limitations chemotherapeutic drugs is that many drugs cannot prevent tumor recurrence after drug withdrawal [149]. For example, tumor recurrence was reported after DOX treatment [150,151]. DOX-loaded self-healing hydrogel composed of chondroitin sulfate multialdehyde, branched polyethylenimine, and graphene showed higher anticancer activity compared to free DOX. Additionally, in presence of a near-infrared laser (NIR)-triggered photothermal effect, the DOX-loaded hydrogels reduced the recurrence of breast cancer cells by about 33.3% compared to free DOX. It should be noted that the hydrogel exhibited about 100% self-healing efficiency along with improved mechanical strength (7000 Pa). These potential findings indicated that SHH could be used to prevent tumor recurrence. Similarly, supramolecular hydrogel (isoGBG) synthesized using isoguanosine, borate and guanosine exhibited biocompatibility, anticancer activity, and self-healing properties. isoGBG hydrogel decreased the viability of cancer cells via apoptosis and also prevented tumor relapse [152].

SHH can be used to enhance sustained release effect(s) of drugs, drug candidates, or other biomolecules [129,153]. SHH could be used to balance the hydrophobic and hydrophilic features of therapeutic agents [154,155]. For example, SHH prepared using ethylenediaminetetraacetic acid (EDTA), Fe^3+^ and HA exhibited excellent antimicrobial activity against E. coli and S. aureus bacteria. Complexation of Fe^3+^ with EDTA and HA assured sustained release of Fe^3+^, and thereby showed antibacterial activity. More importantly, the Fe^3+^ trapped EDTA-HA based hydrogel showed notable biocompatibility against L-929 mouse fibroblast cells and significantly increased wound healing of female C57BL/6 mice compared to control. Furthermore, topically applied hydrogel for 10 days caused inflammation, inhibited growth of S. aureus, and accelerated cutaneous regeneration (Figure 9) [156]. Similarly, SHH could be used to control the release of anticancer agents. Pandit et al. [157] reported that N,O-carboxymethyl chitosan and multialdehyde guar gum (CGG) based injectable SHH showed excellent mechanical properties (with about 1625 Pa storage modulus) along with biocompatibility against HEK-293 human embryonic kidney cells and red blood cells. Interestingly, the swelling behavior of the CGG hydrogels could be controlled by changing the pH. DOX-loaded CGG hydrogel exhibited different drug release kinetics with the change of pH. For instance, at physiological pH (pH 7.4) CGG hydrogel released 32.13% DOX while at tumoral microenvironment (pH 5.5) it released 67.06% DOX. Furthermore, DOX-CGG hydrogel displayed significant cytotoxicity (~72.13%) towards MCF-7 breast cancer cells [157].

## 4. Conclusions and Future Perspectives

SHH are being shown as promising alternatives in tissue engineering and drug delivery due to their compatibility with natural tissues. One of the major challenges of SHH development is to identify a suitable combination of viscous and elastic properties. Recently, significant progress has been achieved in improving the mechanical characteristics and other properties of SHH for advanced applications in biomedical engineering, including tissue engineering and drug delivery [158]. However, several obstacles still need to be overcome to assure practical acceptability of hydrogels for their biomedical use. One of the major challenges of SHH development is to identify a suitable combination of viscous and elastic properties.

Lipophilicity of hydrogel is a prime concern of SHH for their usage in drug delivery. Hydrogel is a hydrophilic polymer that composts with hydrophilic domains and many pores [3]. In the case of a hydrophobic drug, the loading quantity and the homogeneity of the drug inside the gel matrix may be limited [159]. Additionally, due to high-water content and porous structure, hydrophilic drugs could show a rapid release kinetics [160]. That may cause serious side effects in patients.

Another crucial factor is stability of SHH in the in vivo condition. During the swelling time, most SHH become unstable and disintegrate in water quickly, particularly the physically crosslinked hydrogels [161,162]. Even in a water medium, most physically crosslinked SHH cannot perform the self-healing process [30]. Nonetheless, as an alternative to body organs or tissue, self-healing hydrogel must be able to swell and retain water for a long time [12,163]. Whether the chemically crosslinked hydrogels are mechanically tough and stable in water, they provide a very nominal self-healing percentage, which is not significant for real-time biomedical engineering applications [16]. Another major challenge is the trade-off relationship between mechanical toughness, self-healing, and biocompatibility, seriously reducing hydrogel’s biomedical applications. 

There are also ongoing difficulties in extending the kinetic release profiles obtained by employing hydrogels as the drug delivery system [159]. Increasing the release time would be useful, and hydrogels could be used instead of hydrophobic micelle systems for long-term release applications if the release time could be longer [164]. This would be beneficial because of the better biocompatibility of hydrogels. There is a possibility that hydrogels with varying degradation profiles and/or environmentally sensitive segments could be of assistance in resolving these kinetic difficulties [165,166].

Moreover, improvement in mechanical properties is necessary for SHH [3]. Mechanically unstable gel may increase the chance of toxic effects on the body. Mechanically tough and self-healing gels decrease discomfort and suffering while keeping the wound in its natural condition and increasing the dressing material’s lifespan [167]. Therefore, mechanical properties of the SHH need to be tuned for their applications in biomedical applications including drug delivery systems, tissue engineering, and wound healing. Lastly, before clinical trial(s) of SSH, their therapeutic potential, long-term safety, and potential risks should be examined.

## Figures and Tables

**Figure 1 polymers-14-04539-f001:**
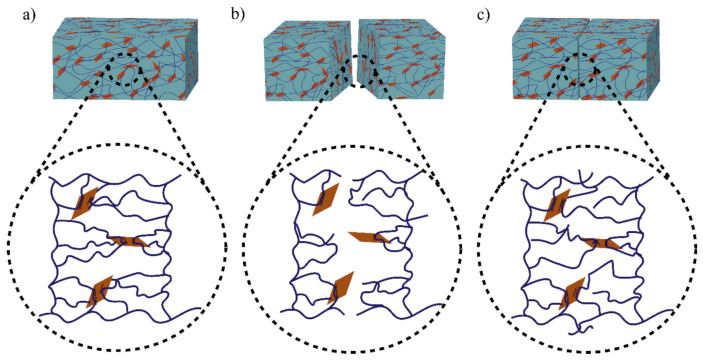
Illustration of the healing mechanism of functionalized graphene oxide crosslinked polyacrylamide hydrogel. Bonding mechanism of (**a**) fresh hydrogel sample, (**b**) cut sample, and (**c**) healed sample. The polymer chains (line) undergoe reversible dynamic hydrogen bonding with the polar OH groups of GO surfaces (rectangle shape).

**Figure 2 polymers-14-04539-f002:**
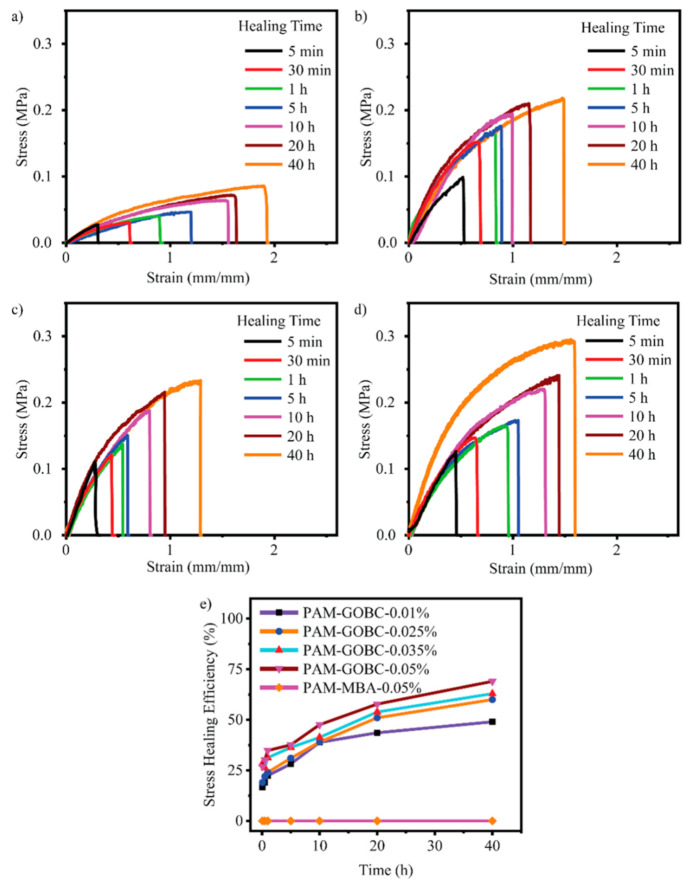
Illustration of the stress-strain curve of healed PAM hydrogels with a (**a**) 0.01%, (**b**) 0.025%, (**c**) 0.035%, (**d**) 0.05% GOBC, and (**e**) demonstrated the stress healing kinetics with different time. This picture adopted from Rumon et al. [16]. “This study is under Creative Commons Attribution 4.0 International License, which permits use, sharing, adaptation, distribution, and reproduction in any medium or format, as long as you give appropriate credit to the original author(s) and the source as well as provide a link to the Creative Commons license (http://creativecommons.org/licenses/by/4.0/, accessed on 19 October 2022)”.

**Figure 3 polymers-14-04539-f003:**
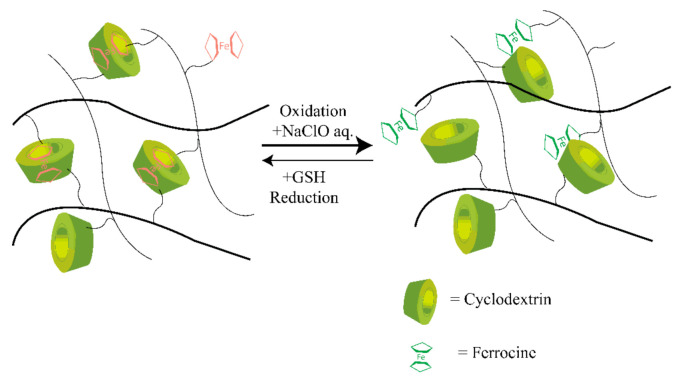
Illustration of the mechanism of host–guest interaction-based redox-active SHH. CD undergoes host-guest interaction with Fe ions through redox phenomena and which results in the self-healing properties. Self-healing hydrogel: SHH; cyclodextrin: CD. CDs undergo the desired bonding with the ferrocene molecules marked as the red color, while on the right side, host–guest bonds are broken in the presence of an oxidizing agent.

**Figure 4 polymers-14-04539-f004:**
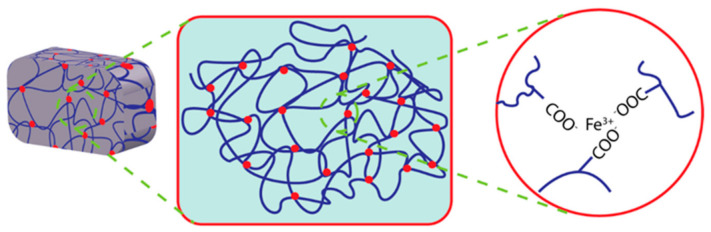
Illustration of Fe ion coordinated self-healing hydrogel.

**Figure 5 polymers-14-04539-f005:**
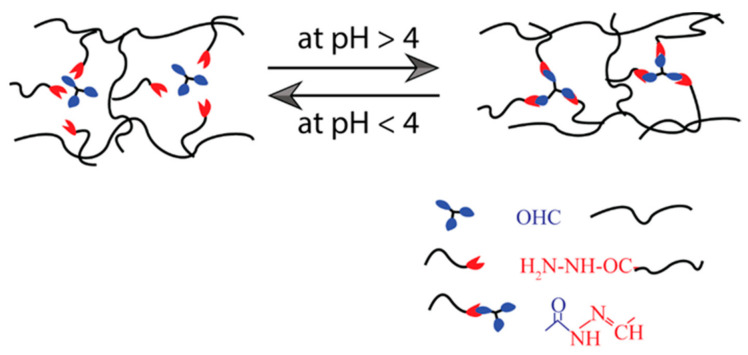
Illustration of self-healing mechanism of pH-dependent Schiff base bond-based hydrogel.

**Figure 6 polymers-14-04539-f006:**
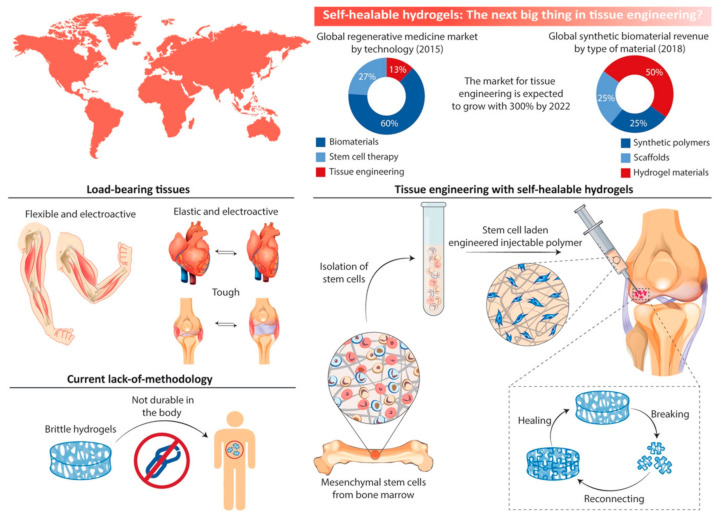
Applications of self-healing hydrogel in various tissue engineering applications. This figure is adopted form Talebian et al. [12]. “This study is under Creative Commons Attribution 4.0 International License, which permits use, sharing, adaptation, distribution, and reproduction in any medium or format, as long as you give appropriate credit to the original author(s) and the source as well as provide a link to the Creative Commons license (http://creativecommons.org/licenses/by/4.0/, accessed on 19 October 2022)”.

**Figure 7 polymers-14-04539-f007:**
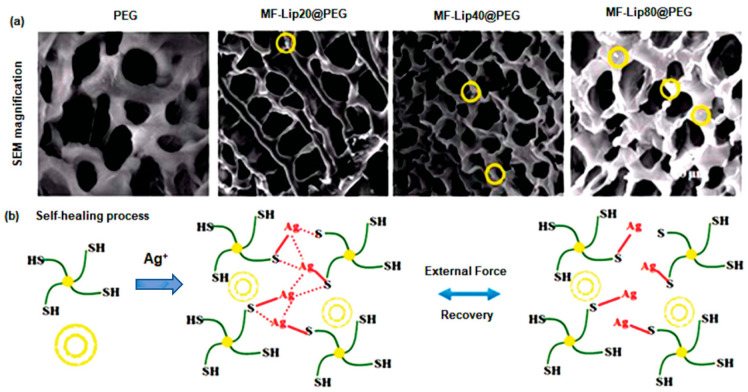
Microscopic examination and special features of MF-Lip@PEG hydrogels. (**a**) SEM images (yellow circles indicate liposomes). (**b**) Self-healing process in vitro drug release kinetics. This figure is adapted from Mao et al. [135]. Yellow bilayer circle indicates liposome and green marked line connected with solid yellow circle indicates PEG.“This study is under Creative Commons Attribution 4.0 International License, which permits use, sharing, adaptation, distribution, and reproduction in any medium or format, as long as you give appropriate credit to the original author(s) and the source as well as provide a link to the Creative Commons license (http://creativecommons.org/licenses/by/4.0/, accessed on 19 October 2022)”. MF-Lip: Mangiferin liposome; PEG: Polyethylene glycol.

**Figure 8 polymers-14-04539-f008:**
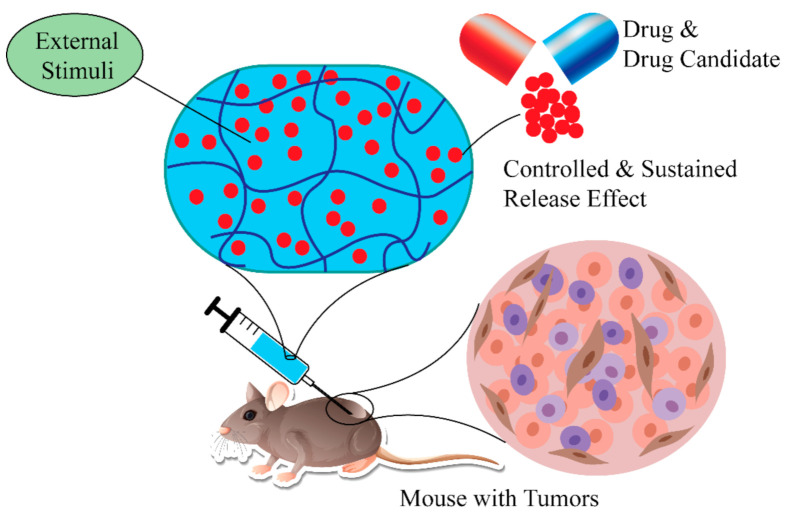
Using SHH in delivery of therapeutic agents. SHH effectively delivery therapeutic agents to the target tissue. Furthermore, drug release kinetics can be controlled by external stimuli (temperature, pressure, magnetic field). SHH: Self-healing hydrogels.

**Figure 9 polymers-14-04539-f009:**
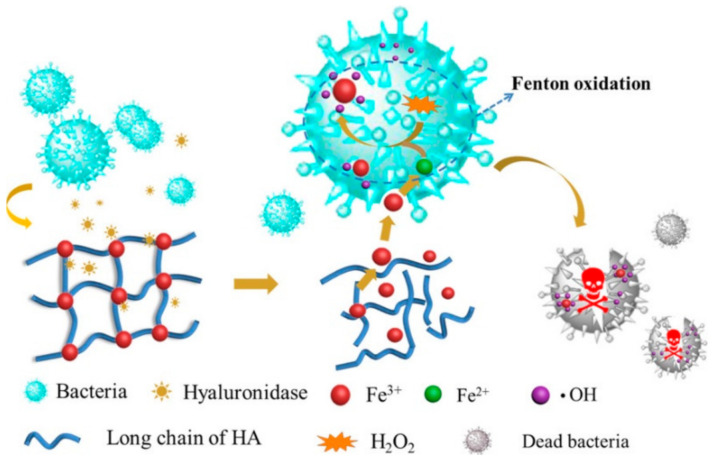
Sustained release effects of Fe^3+^ loaded SHH. SHH: Fe^3+^ -HA complex was rapidly adsorbed by bacteria and subsequently caused the reduction of Fe^3+^ to Fe^2+^. Fe^2+^ reacts with H_2_O_2_ to generate hydroxyl radical (.OH) which damages the bacterial proteins or nuclear acids. This figure is reprinted from Tian et al. [156]. “This study is under Creative Commons Attribution 4.0 International License, which permits use, sharing, adaptation, distribution, and reproduction in any medium or format, as long as you give appropriate credit to the original author(s) and the source as well as provide a link to the Creative Commons license (http://creativecommons.org/licenses/by/4.0/, accessed on 26 September 2022)”. HA: Hyaluronic acid; ·OH: Hydroxyl radical; SHH: Self-healing hydrogel.

**Table 1 polymers-14-04539-t001:** Comparison of self-healing efficiency of various hydrogels prepared through different strategies.

Mechanism	Polymer/Materials	Healing Condition	Self-Healing Efficacy	Ref.
Host–guest	β-functionalized AM and N-AD functionalized-AM copolymer	24 h, 20 °C	24 h, 74%,	[56]
Host–guest	β-CD and α-bromonaphthalene-functionalized AM with 6-thio-CD-modified AuNPs	1–60 min, 20 °C	≈100%, 1 h	[57]
Host–guest	β-CD, AD, and Fc	2 h, 20 °C	68%, 25 °C	[58]
Host–guest	AA-6β-CDs and AA-Fc	24 °C	24 h, 84% h	[59]
Hydrophobic	CTAB, SDS, and NaCl Am-*co*- SMA	3 min, 20 °C	8%, 60 min, (modulus)	[60]
Hydrophobic	NIPAM, SMA, CTAB and NaBr	30 min, 24 °C	100%, 30 min	[61]
Hydrophobic	SDS and AM-*co*-OPPEA octylphenol polyethoxyether acrylate copolymer	3 days, 25 °C	After 3 d ≈40%, and ≈70%, stress at break respectively	[62]
Hydrophobic	AM-*co*-C18 in SDS and NaCl solution	20 °C		[52]
Hydrophobic	AM-*co*-C18 in SDS and NaCl solution	24 °C, 10 min	100%, 20 min,	[63]
Hydrophobic	Am-*co*-SMA in presence of SDS and NaCl solution		29%, 34%, and 20% for acrylates of C16, C18, and C22, respectively	[64]
Hydrogen Bonds	PVA-*g*-DOPA	25 °C	100% within 270 s	[65]
Hydrogen Bonds	PVA	25 °C	48 h, 72% healing	[44]
Hydrogen bond	PAM and GOBC	48 h, 25 °C	70%	[16]
Hydrophobic bonds	SMA -*co*- Am in SDS-NaCl solutions	20 °C	100% within 1 s	[52]
Hydrophobic bonds	SMA copolymerization with Am in CTAB, SDS, and NaCl solutions	35 °C	98%	[60]
Hydrophobic bonds	Noctadecyl acrylate and AA	80 °C	87% healing in 24 h	[66]
Hydrogen Bonds	PUPEGMA-*co*-UPy	At ambient RT	Within 5 min, >60% healing	[67]
Ionic bonds	=AA with FeCl_3_ polymerization	25 °C	6 h, 66%	[68]
Ionic bonds	MPTC and NaSS	RT	12 h, 66%	[69]

Stearyl methacrylate (C18): SMA; Poly(ethylene glycol) methacrylate end-capped urethane ether prepolymer: PUPEGMA; Ureido pyrimidinone: Upy; Acrylic acid: AA; Acrylamide: AM; Polyvinyl alcohol: PVA; Cetyltrimethylammonium bromide: CTAB; Sodium dodecyl sulfate: SDS; 3- (methacryloylamino) propyltrimethylammonium chloride: MPTC; Sodium pstyrenesulfonate polymerization in two steps: NaSS; *β*-cyclodextrin: *β*-CD; Ferrocene: Fc; Graphene oxide based crosslinker: GOBC; adamantine: AD; Octyl phenol polyethoxy ether acrylate copolymer: OPPEA.

**Table 2 polymers-14-04539-t002:** Comparison of self-healing efficiency of hydrogels prepared for 3D/4D printing.

Mechanism	Hydrogels	Reaction Condition	Self-Healing Efficacy	Ref.
Disulfide and acylhydrazone exchange	Dithiodipropionic acid dihydrazide functionalized PEO and HAD	48 h at 25 °C with different pH	50%, 48 h	[93]
Acylhydrazone exchange and Schiff base	*N*-carboxylethyl CA, OSA, and adipic acid dihydrazide	6 h, 25 °C, pH 7	86 ± 3.4%, 12 h and 90 ± 2.7%, 48 h	[94]
Acylhydrazone bonds	PEO and tris[(4-formylphenoxy) methyl] ethane catalyzed by acetic acid.	At ambient RT	7 h	[95]
Schiff base	CA or GCA, and telechelic difunctional PEG	2 h, RT (20 °C)	≈51%, at the start of healing and ≈91% after 10 min of healing	[86]
Schiff base	AMC and OA	1–3 h, 20 °C		[96]
Schiff base	CSMA and SC			[97]
Schiff-base	CMBC-EDTA and CHO-Pectin		23% at pH = 7	[98]
Schiff-base	CA or GCA, and DF-PEG.	15 min, 30 min, 20 °C	≈390 Pa at 0 min healing time and ≈2000 Pa for virgin sample	[99]
Disulfide bonds	HAuCl_4_/AgNO_3_ solutions and tetra-arm thiol-terminated PEG homopolymer [(PEGSH)_4_]		90% mechanical healing	[100]
Oxime bonds	P(DMA0.68–stat–DAA)	25 °C	2 h	[101]
Imine bond	GCA and DF-PEG		Healed Within 15 min	[99]
Imine bond	PLGA-g-PCL		81% within 6 h	[102]
Boronate ester bonds	Borax, PEGDA, DTT and PEG	25 °C	Healed within 30 min	[103]
Diels-Alder	Dex-l-PEG and PEG–DiCMA	37 °C	7 h, 98.7% healing	[104]

Hex-anedioic acid dihydrazide: HAD; Chitosan: CA; glycol chitosan: GCA; and telechelic difunctional poly(ethylene glycol): PEG; oxidized sodium alginate: OSA; 2-acrylami-dophenylboronic acid: ADBA; *N,N′*-dimethylacrylamide: DMA; poly(vinyl alcohol): PVA; dopamine acrylamide: DOPAA; poly(*N,N*–dimethylacrylamide-stat-diacetone acrylamide): P(DMA–stat–DAA); poly(ethylene oxide): PEO; dextran-l-poly(ethylene glycol): Dex-l-PEG; dichloromaleic-acid-modified poly(ethylene glycol): PEG-DiCMA, carboxyl methyl bacterial cellulose: CMBC.

## Data Availability

Data are available in this article. There is no supporting document.

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
