# Peer review of "Self-Healing Hydrogels: Development, Biomedical Applications, and Challenges"

_polymers, 2022, doi:10.3390/polym14214539_

Round 1

Author Response

Reviewer 1:

Comments and Suggestions for Authors

The authors presented a review topic regarding Self-healing Hydrogels: Development, Biomedical Applications, and Challenges, which interested readers. Polymeric hydrogels are biological materials with similar mechanical and chemical properties to natural tissues. Self-healing hydrogels (SHH) show potential in tissue engineering and medication delivery. External cues can control their responses (e.g., pH, temperature, pressure, radiation). Developing SHH requires a combination of viscous and elastic materials, and real-time applications were constrained by the trade-off between healing performance and mechanical robustness. This study discusses SHH mechanisms, current advances, tissue engineering, and medication delivery problems. The reviewer concluded that the paper has potential and could be published. Please revise based on the reviewer's comments.

General comments

In general, this manuscript is well-written, and it contributes to the body of knowledge. The subject matter is presented in a comprehensive manner, and the references are appropriate for the topic being discussed.

Response: Thanks for your time and carefully reviewing the manuscript of Rumon et al. Here is the point by point responses of your comments.

Point 1: Please combine Section 2 and Section 3 into one section because Section 2 only has

three sentences (Lines 99-103).

Response 1: We have marge the section 2 and 3 into one section. Please see the line number 107-331 (in the pdf version). Thank you.

Point 2: Figures 1, 2, 3, 4, 5: Please cite the source of the figure if the authors do not make the results. Also, please provide proof of the author's permission for REUSING the figure.

Response 2: Figures 1, 2, 3, 4, 5 are illustrate by our author. No need to take permission for these figures. Thank you

Point 3: Figure 5: There are 2 Figures 5. Please revise.

Response 3: We are sorry for the typo. In the revised manuscript figures are organized according to order. Thanks

Point 4: Please mention or explain Figure 4 in the text.

Response 4: Figure 4 (Figure 5 in the revised manuscript) has been mentioned in the text. Please see the line number 268 (in the pdf version). Thank you.

Point 5: Please combine Sections 5 and 6 into one section (Conclusions)

Response 5: We have marge the section 5 and section 6 into one section entitled as Conclusion and Future Perspectives. Please see the line number 514-554 (in the pdf version). Thank you.

Point 6: References: Please check the ref. [1]. There is a parenthesis at the first of the row.

Response 6: Reference 1 has been checked and provided in correct form. Please see the line number 556. Thank you.

Point 7: References: Line 781: please remove the stars symbol “*”.

Response 7: the stars symbol “*” has been removed. Please see the reference 150. Thank you.

  1. Questions:

2.1 What sources of references did you choose for this study? Web of Science or Scopus Data or CNKI and Google Scholar? And then please explain why you choose those sources.

Answer 2.1: We have added article searching strategy in the introduction part. Please see the line number 95-101 (in the pdf version)

2.2 Please rewrite the abstract and the conclusions (new conclusions from already published works) to remark on significant stages of the field development. Besides, please identify the research gaps and explore potential areas in a particular field.

Answer 2.2: Abstract and conclusion have been revised according to your valuable suggestion. Thanks

Reviewer 2 Report

The Authors wanted to present a review of self-healing hydrogels. Nevertheless, I m not fully convinced that all of the presented systems can be named hydrogels. A major revision is needed.

The strong side of the manuscript is the relatively current status of the described research achievements - most of them were published 3-5 years ago (older cited suitably where needed). However, in the manuscript there is a shortcut - please check the detailed comments below.

The possible biomedical application focused mainly on drug delivery. What about wound healing, etc.?

A major revision is necessary.

Abstract

- please define shortly here what you mean by SHH.

Introduction

line 35 please check the grammar of this sentence

line 52 please explain what you mean by “The tricky job associated…”

line 61 "The modulus..." - what kind of modulus

line 65 "SHH have large viscous properties" - what does it mean?

line 75 - this section in my opinion does not describe hydrogels. How solid particles might affect the SHH structure - please check it carefully

Paragraph 2 can be cross-out and inserted partially in paragraph 3

line 125 - please check "graphene oxide-based polyacrylamide" is that hydrogel?

line 133 AA-DDD - define and present in the graph

Table 2 what you mean by "Reaction condition"

line 313 SHH is one the most promising - this general statement was presented several times with different references. Please add more examples than general statements.

figure or table which will summarize potential applications of SHH is strongly recommended in this section.

figure 5 is not meaningful. Now, it looks like you need several factors together which should be applied for SHH if their work is needed. Please change it.

line 389 - where is self-healing mechanism here?

Author Response

Reviewer 2: 
Comments: 
The Authors wanted to present a review of self-healing hydrogels. Nevertheless, I’m not fully 
convinced that all of the presented systems can be named hydrogels. A major revision is needed.
The strong side of the manuscript is the relatively current status of the described research 
achievements - most of them were published 3-5 years ago (older cited suitably where needed). 
However, in the manuscript there is a shortcut - please check the detailed comments below.
Response: Thanks for your time and carefully reviewing the manuscript of Rumon et al. Your 
major comments help to improve the quality of our article. Here is the point by point responses of 
your comments.
Point 1: The possible biomedical application focused mainly on drug delivery. What about wound 
healing, etc.?
Response 1: There are many potential biomedical applications of self-healing hydrogels. 
Previously, we focused only tissue engineering and drug delivery. As per your valuable suggession 
we added wound treatment section (section 3.2). Thank you. 
Abstract
Point 2: - Please define shortly here what you mean by SHH.
Response 2: We have defined the short form of SHH in introduction. Please see the line number 
15 (in the pdf version). Thank you.
Introduction
Point 3: line 35 please check the grammar of this sentence
Response 3: We have checked it and rewritten with correct grammatical form. Please see the line 
number 37-39 (in the pdf version). Thank you.
Point 4: line 52 please explain what you mean by “The tricky job associated…”
Response 4: We have defined and explained the term ‘The tricky job’ in the very previous 
paragraph. We were trying to say about the trade-off relation among the mechanical toughness and 
the self-healing behavior. The Please see the line number 47-52. Thank you.
Point 5: line 61 "The modulus..." - what kind of modulus
Response 5: We have replaced the word modulus with Young's modulus. Please see the line 
number 63. Thank you.
Point 6: line 65 "SHH have large viscous properties" - what does it mean?
Response 6: We have rewritten this part. Here, we indicated that viscous properties facilitate the 
self-healing properties through increasing the molecular chain diffusion. Please see the line 
number 67. Thank you.
Point 7: line 75 - this section in my opinion does not describe hydrogels. How solid particles might 
affect the SHH structure - please check it carefully
Response 7: Functionalized nanoparticles might be facilitating the self-healing properties. There 
are several functional nanoparticles including graphene oxide, carbon nanotube, ellulose etc. to 
introduce the healing properties. Please see the line number 77-79. Thank you.
Point 8: Paragraph 2 can be cross-out and inserted partially in paragraph 3
Response 8: We have marge the section 2 and section 3 into one section. Please see the line 
number 107-331 (in the pdf version). Thank you.
Point 9: line 125 - please check "graphene oxide-based polyacrylamide" is that hydrogel?
Response 9: Yes, this is a hydrogel and published by one of our authors in RSC advances. Please 
see the reference number 16. Thank you.
Point 10: line 133 AA-DDD - define and present in the graph
Response 10: We have explained it. Please see the line number 155-156. Thank you.
Point 11: Table 2 what you mean by "Reaction condition"
Response 11: Table 2, "Reaction condition" is replaced by healing condition. Please see the line 
number 270. Thank you.
Point 12: line 313 SHH is one the most promising - this general statement was presented several 
times with different references. Please add more examples than general statements. Figure or table 
which will summarize potential applications of SHH is strongly recommended in this section.
Response 12: We have added a figure as a part of main manuscript. Please see the figure number 
6. Thank you.
Point 13: figure 5 is not meaningful. Now, it looks like you need several factors together which 
should be applied for SHH if their work is needed. Please change it.
Response 13: We have re-illustrate the figure 5. Please see the current figure 8. Thank you.
Point 14: line 389 - where is self-healing mechanism here?
Response 14: We did not report self-healing mechanism here. In this section we mainly focused
on the usage of self-healing hydrogels in drug delivery system, their potency and more. 

Round 2

Reviewer 2 Report

The Authors strongly improved the manuscript content due to my comments.

In my opinion, the manuscript is ready to present in its current form.

Two editorial remarks which do not impact the general manuscript quality:

- Figure 3,4 - in the printed version might be enlarged - the fonts i.e. of Fe will be more visible

- please consider adding one more figure to present how the described hydrogels look in "real life".